# Degradation of chondroitin sulfate A by a PUL-like operon in *Tannerella forsythia*

**Peter Nguyen, Rony Eshaque, Barbara Anne Garland, Anthony Dang**[ID]**, Michael D. L. Suits**[ID]*

Department of Chemistry and Biochemistry, Wilfrid Laurier University, Waterloo, Ontario, Canada

* msuits@wlu.ca

**Data Availability Statement:** The data underlying the results presented in the study are available from the protein data bank (https://www.rcsb.org/). Protein Data Bank: accession numbers 8DI0 and 8DI1 for Bfo2290 and Bfo2294, respectively.

## Abstract

Advanced periodontitis has been shown to have strong association with the residence of the bacterial consortia known as the red complex comprised by *Porphyromonas gingivalis*, *Tannerella forsythia*, and *Treponema denticola*. *T. forsythia* shares a distant genetic linkage to *Bacteroidetes thetaiotaomicron* and may therefore produce analogous polysaccharide utilization loci (PUL) which enable complex carbohydrate degradation, import, and use, although this capacity has yet to be demonstrated. Chondroitin sulfate A is a linear, sulfated carbohydrate linked to periodontal disease as the principal species of glycosaminoglycan appended on the surface of cortical bone of teeth and in supporting dental ligaments. Through genomic comparisons with *B. thetaiotaomicron*, a new PUL-like operon (Bfo2285-Bfo2295, and Bfo3043) was identified in *T. forsythia* and the crystal structure of two proteins from this PUL-like operon, Bfo2290 and Bfo2294, were reported using X-ray crystallography. Enzyme kinetics for Bfo2290 were reported using a pH-dependent assay and suggested a $K_m$ of 0.75 mg/ml ± 0.60 mg/ml, $K_{cat}$ of 3.74 min$^{-1}$ ± 0.88 min$^{-1}$, and $V_{max}$ of 7.48 μM/min ± 1.76 μM/min with partially degraded chondroitin sulfate A. Fluorophore-assisted carbohydrate electrophoresis was used to show the processive degradation of chondroitin sulfate A by the proteins encoded in *T. forsythia* PUL-like operon, and revealed Bfo2291 and Bfo2290 to be an endolytic chondroitin sulfate A lyase and exolytic ΔDi-4S chondroitin sulfate A sulfatase, respectively.

## Introduction

The human oral cavity harbours a species-rich microbiota of greater than 700 bacterial species, the majority of which are embedded in extracellular polymeric substances, such as dental plaque biofilms [1]. Microorganisms that inhabit the oral cavity typically form well-defined consortia, with species sharing resources and constructing biofilm matrix comprised by polysaccharides, peptides, and extracellular DNA. While in a state of homeostasis, these oral biofilms prevent the colonization of exogenous species through the production of inhibitory agents such as bacteriocins, organic acids, hydrogen peroxide, and antibacterial enzymes [2, 3]. However, at a complex intersection of host oral health, genetic and environmental factors, and microbe-immunological response, the excessive buildup of oral biofilms, or the residence

**Funding:** PN, BG, RE, and AD were supported by studentships provided by Wilfrid Laurier University. This work was supported by a National Science and Engineering Research Council of Canada Discovery Development Grants (RPGIN-2014-05018 and DDG-2020-00007). Additional support was provided by GlycoNet Grant AM-21. Synchrotron data were collected at the Canadian Light Source, Cornell High Energy Synchrotron Source, and infrastructure supported by the Canadian Foundation for Innovation, the Province of Ontario, and Laurier. The funders had no role in study design, data collection and analysis, decision to publish, or preparation of the manuscript.

**Competing interests:** The authors have declared that no competing interests exist.

of pathogen consortia can result in a breakdown of homeostasis and a change in microbial composition to an eventual chronic state of dysbiosis [4]. Periodontal disease is a chronic inflammatory disease of periodontal tissue caused by this dysbiosis [2]. Symptoms of periodontal disease can be as mild as halitosis, swelling and redness of periodontal tissue, to purulent discharge, bone loss, and tissue necrosis in cases of severe periodontitis [5]. Recent advances in cataloguing and characterization of the human oral microbiome have highlighted many species loosely associated to be the cause of periodontal disease, but many of the molecular mechanisms remain undetermined.

In juvenile patients with clinical periodontitis, the microbial population of subgingival biofilms typically have an increased presence of obligatory anaerobic asaccharolytic gram-negative organisms and a decreased presence of beneficial gram-positive organisms which produce the previously mentioned beneficial inhibitory agents [6, 7]. The presence of a consortia of bacteria known as the red complex comprised by *Treponema denticola*, *Porphyromonas gingivalis*, *and Tannerella forsythia*, have been established as a hallmark sign of periodontitis across all demographics [8, 9]. The red complex organisms are highly dependent on each other for proliferation and virulence [9]. The deployment of bacterial adhesion allows *T. forsythia* and *P. gingivalis* to be effectively transported through a "piggyback" mechanism by attaching themselves to *T. denticola*; the only motile species of the red complex [10, 11]. Red complex bacteria also adhere to host epithelial cells not only to secure themselves and form biofilms, but to associate and then penetrate host cells by activation of host signalling pathways to protect it from host immune responses [12–15]. Once established, red complex bacteria disrupt host tissue by inducing host cell apoptosis, producing proteolytic enzymes, and by stimulating host inflammation and defence responses, which cause the symptoms associated with periodontal disease such as bleeding gums and tooth destabilization [16–21].

Proteoglycans are cell surface structures composed of complex carbohydrates and an anchoring protein that interact with a variety of other substances and structures to serve different biological processes through wide-ranging chemical and biophysical properties [22]. Glycosaminoglycans (GAGs) are a carbohydrate component of proteoglycans and are found appended to proteoglycans of connective tissues such as periodontal tissue [23]. GAGs are composed of repeating disaccharide subunits typically consisting of an N-acetylated sugar (D-glucosamine or D-galactosamine), and a uronic acid (D-glucuronic acid or L-iduronic acid) [22]. In cortical bone, the GAG chondroitin sulfate A (CSA), composed of repeating D-glucuronic acid and N-acetylgalactosamine moieties connected by alternating β-1,4 and β-1,3 glycosidic bonds, and sulfated at C4 of GalNAc, is the principal GAG species. CSA is thought to be the most abundant at this site due to its distinct spatial interactions with collagens which are responsible for the mineralization of bone matrices [24]. The corollary is that a disruption in CSA may disrupt bone mineralization; leading to a loss of supporting bone structure which is a distinguishing feature of severe periodontitis [5].

Due to the distant genetic lineage of *T. forsythia* to *Bacteroidetes thetaiotaomicron*, *T. forsythia* may also contain PUL-like gene clusters which are well-characterized in the latter species. *B. thetaiotaomicron* has been extensively studied and found to be a predominant species in human gut-microbiomes [25]. *B. thetaiotaomicron* is recognized as having diverse metabolic capability, and has a genome that encodes a high number of gene clusters that express proteins for utilizing a large range of complex glycans [26]. These gene clusters encode for multi-component protein systems tailored for detection, binding, transporting and degrading specific carbohydrates [27]. Hallmark features of PULs are tandem <u>s</u>tarch-<u>u</u>tilization <u>s</u>ystem (*Sus*)C and *SusD* genes which encode for a TonB-dependent transporter and a carbohydrate-binding lipoprotein respectively [25]. While *B. thetaiotaomicron* may have diverged with evolutionary pressure in the gut towards metabolic diversity, other species that emerged from a common

evolutionary ancestor, such as *Parabacteroides distasonis* and *Parabacteroides merdae*, have genomes that encode for fewer and more specialized metabolic functions that specifically target host tissues [28]. While these three organisms are all considered gut-associated microbes, *P. distasonis* and *P. merdae* encode the Sus locus structure in their genome and are genetically much closer to *T. forsythia* than *B. thetaiotaomicron*, suggesting that *T. forsythia* may have a comparable strategy for carbohydrate-utilization, deploying sPUL-like Sus gene loci that functionally target specific host glycans [28].

Alongside increased osteoclastogenesis by host cells [29], bone loss in patients with severe periodontitis may also be attributed to the degradation of GAGs by PUL-like gene clusters found in organisms such as *T. forsythia*. It has been established that the destruction of periodontal tissue is attributed to host immune responses triggered by periodontal pathogens as well as directly through tissue disrupting virulence factors [11, 18, 19, 30–35]. Along with the previously mentioned proteolytic enzymes, glycosidases have also been recognized as potential virulence factors produced by red complex bacteria [8]. For example, two sialidases SiaH and NahH are expressed by *T. forsythia* which catalyze the cleavage of terminal sialic acids from complex carbohydrates of glycoproteins and glycolipids [31–35]. At least 8 other glycosidases have been identified in *T. forsythia*, and in order to decipher the role these glycosidases play in contributing to pathology of periodontal disease, isolation and biochemical characterization should be undertaken [36].

Here we define a pathway for the degradation of CSA by *T. forsythia* based on the discovery of a PUL-like gene cluster found using bioinformatics analysis, functionally characterize three proteins (Bfo2285, Bfo2290, Bfo2291, and Bfo2294) from this PUL-like gene cluster and report the crystal structure of an exo-*N*-acetylgalactosamine (Bfo2290) and for a 2-keto-3-deoxy-6-phosphogluconate (KDPG) and 2-keto-4-hydroxyglutarate (KHP) to D-glyceraldehyde-3-phosphate aldolase (Bfo2294). These four proteins were functionally characterized to better understand the degradation of CSA by this pathway at the biochemical level. The activity of these proteins on CSA may suggest a mechanism by which *T. forsythia* directly degrades bone tissue and destabilize the dental ligaments in patients with chronic periodontal disease.

## Methods and materials

### Bioinformatics analysis and pathway modelling

Bioinformatics analysis and pathway modelling was completed using *T. forsythia* strain 92A2 (GeneBank: CP003191.1) and a recently characterized PUL from *B. thetaiotaomicron* [26]. The genome of *T. forsythia* was searched using a basic local alignment search tool (BLAST) from the National Center for Biotechnology Information (NCBI) for *SusC* and *SusD* gene pairs near uncharacterized putative glycosidases [37, 38]. Once *SusC* and *SusD* gene pairs were found, upstream and downstream proteins were analyzed using FGENESB and DOOR2 in order to determine the boundaries of the operon using *B. thetaiotaomicron* strain VPI-5482 (GeneBank: AE015928.1) as a model [39, 40]. Cellular localization was predicted using PSORTb v.3.0.2 and SignalP-5.0 [41, 42]. Putative function prediction, and homology modelling was conducted using InterPro, Phyre2, and Swiss-Model respectively [43–45]. The localization prediction and function prediction of proteins within the predicted operon boundaries were compared to the PUL described by Ndeh *et al.*, then missing proteins were searched for using STRINGv11 to predict protein-protein interaction networks [26, 46]. Using the predicted pathway model, the first three glycosidases (*Bfo2291*, *Bfo2290*, and *Bfo2285* respectively) were chosen as targets for recombinant expression. ExPASy ProtParam and I-TASSER were used to predict biophysical characteristics for purification purposes and for cofactors and ligand binding respectively [47, 48].

The three sequences selected for recombinant expression (Bfo2291, Bfo2290, and Bfo2285) were truncated prior to synthesis and transformation. Truncation boundaries were guided by InterPro domain predictions using full length sequences obtained from UniProt [43, 49]. The full-length Bfo2285 protein is 405 amino acids long and was truncated to residues 36–405 to remove a single peptide and a prokaryote membrane lipoprotein lipid attachment site from residues 1–21 and 1–19 respectively. Full-length Bfo2290 is 516 amino acids long. Residues 1–21 were predicted to contain a signal peptide and the gene was truncated to encompass residues 31–512, the predicted sulfatase-like domain. The full-length sequence of Bfo2291 is 641 amino acids long and was truncated to residues 30–581 which contained a chondroitin AC lyase like domain from residues 30–391 and a heparinase II/III-like domain from residues 410 to 581. The N-terminal truncated amino acids were predicted to contain three different signal peptide regions from residues 1–20. The truncated sequences used for recombinant expression can be found in S1 Table.

## Recombinant expression and purification of Bfo2285, Bfo2290(+), Bfo2291, and Bfo2294

Codon-optimized genes were synthesized and inserted into expression vectors by BioBasic and transformed as previously described [50]. Bfo2291 and Bfo2294 were transformed using pET28a plasmids into *Escherichia coli* BL21 (DE3) cells. Bfo2290 was transformed using pET28a plasmids into Rosetta *E. coli* cells as well as in a pET-Duet plasmid that also contained an arylsulfatase maturase enzyme (Bfo1635) which was co-expressed to produce catalytically active protein, henceforth referred to as Bfo2290+. Bfo2285 was expressed using a pET28a plasmid constructed using a PCR amplified gene from pET21b plasmids into Rosetta *E. coli* cells. Bfo2285 was amplified using PCR from a stock of pET21b plasmids containing the gene that was miniprepped using the Qiagen miniprep kit from a previously transformed stock of *E. coli* DH5α cells. The PCR mixture contained 1 μl of both the reverse and forward flaking primer from a 20 μM stock, 5 μl of template DNA at 4 ng/ml, 5 μl of 10x NEB PCR buffer, 1 μl of 10 mM deoxynucleoside triphosphate (dNTP), 0.5 μl *Taq* polymerase, 2 μl dimethyl sulfoxide (DMSO), and 34.5 μl nuclease free water. The sequence for the forward and reverse flanking primers can be found in S2 Table. 30 cycles were performed using a touchdown gradient annealing approach from 61°C to 51°C at a ramp rate of -0.3°C per cycle. The amplified DNA was extracted using a Qiagen gel extraction kit and digested using both *Nde*I and *Xho*I restriction enzymes. A 10ml culture of NEB5-α cells transformed with empty pET28a plasmid was used for a miniprep and was also digested using *Nde*I and *Xho*I. Ligation of the amplified Bfo2285 gene was accomplished using 1ul of amplified Bfo2285 DNA at a concentration of 300 ng/μl, 10ul of 5x NEB ligase buffer, 1 ul of T4 NEB T4 DNA ligase, 3 μl of digested pET28a plasmid DNA, and 35 μl of nuclease free buffer. The mixture was incubated at 37°C for 1 hour, before inactivating the proteins using a heat block at 80°C for 20 minutes, and then transformed as previously described [50]. The primer sequences can be found in S2 Table.

Successfully transformed cells were used for bacterial growth and overexpression to produce protein. Freezer stocks of newly transformed cells were produced using a mixture 1:1 mixture of 70% (w/v) glycerol and lysogeny broth (LB) containing newly transformed cells and rapidly frozen using liquid nitrogen. Freezer stocks were used to inoculate 10ml cultures containing LB media and 10 ul of kanamycin from a 50 mg/ml stock (an additional 10ul of chloramphenicol was added from a 25 mg/ml stock solution for Rosetta *E. coli* cells). The 10 ml cultures were grown for 16 hours at 37°C in a shaking incubator at 240 rpm, then used to inoculate larger 500 ml cultures containing enriched LB media with twice the concentration of tryptone and yeast extract (2YT) and the same concentration of antibiotics as previously

described. The 500ml cultures were grown at 37˚C in a shaking incubator at 240 rpm until they reached an optical density of 0.6 measured at 600 nm ($OD_{600}$), then induced using isopropyl β-D-1-thiogalactopyranoside (IPTG) to a final concentration of 0.75 mM at 16˚C and at 160 rpm for 16 hours in a shaking incubator. Cells were then collected by centrifuge for 10 minutes at 4400 g, resuspended in lysis buffer (100 mM (4-(2-hydroxyethyl)-1-piperazineethanesulfonic acid) (HEPES) pH 7.5, 20 mM sodium phosphate and 50mM NaCl, and 100mM HEPES pH 7.8 for Bfo2290, Bfo2291, Bfo2294, and Bfo2285 respectively) and lysed using a QSonica Q125 probe-tip sonicator with the Q125 system standard probe at 60% peak amplitude for 10 minutes in 10/20 second on/off cycles on ice. The lysate was clarified by centrifugation at 20,000 g for 45 minutes.

Since a short C-terminal histidine-affinity tag was added to each target prior to being sent for synthesis, IMAC was used for purification of Bfo2291, Bfo2290(+), and Bfo2294. A 10 ml of gravity packed nickel-nitrilotriacetic acid (Ni-NTA) resin in a 50 ml BioRad glass Econo-Column was equilibrated with 40 ml of the lysis buffer. Then, incubated with the lysate on ice using a rocking platform for 45 minutes. The column was slowly filled with the wash buffer consisting of the lysis buffer, 5 mM imidazole, and 300 mM sodium chloride (NaCl) and attached to the pump. A gradient of wash buffer and elution buffer (lysis buffer, 500 mM imidazole, and 300 mM NaCl) was used to elute the protein, and 10ml fractions were collected, each with 25 mM stepwise increasing concentrations of imidazole. Fractions were checked using sodium dodecyl sulfate polyacrylamide gel electrophoresis (SDS-PAGE) as previously described, then concentrated to a final volume of 10ml using 30 kDa molecular weight cutoff Millipore Centricon [51]. The concentrated protein was desalted using a General Electric ÄKTA Pure FPLC via two 5 ml HiTrap desalting columns were connected in tandem to increase the column volume and allow for a larger injection volume. The protein was concentrated again to a final concentration of 20mg/ml, measured using $A_{280}$ and the extinction coefficient calculated by ExPASy ProtParam [48].

Ammonium sulfate precipitation was used for purification of Bfo2285. 48ml of saturated ammonium sulfate solution was added dropwise to the clarified lysate, then the lysate was centrifuged at 20,000g for 15 minutes. An additional 30ml of saturated ammonium sulfate solution was added to precipitate Bfo2285, and then the protein was collected using centrifugation at 20,000g for 15 minutes and resuspended in the lysis buffer. The protein was desalted using 50 mm Fisher Regenerated Cellulose dialysis tubing in 2L of lysis buffer for 1 hour at 4˚C, followed overnight dialysis in 2L of fresh lysis buffer at 4˚C. Secondary purification with anion exchange chromatography was accomplished using the ÄKTA Pure FPLC via a GE HiTrap Q FF column and fine gradient elution up to a final concentration of 1M NaCl. The protein was then concentrated to a final concentration of 20mg/ml.

## Fluorophore-assisted carbohydrate electrophoresis

Analysis of CSA (Sigma Aldrich C9819) and enzyme degraded products derived from CSA was analyzed using FACE as previously described [52]. CSA at a concentration of 10 mg/ml was enzymatically digested in 20 mM sodium phosphate pH 7.8 and 50 mM NaCl for about 48 hours at 37˚C. The protein digest matrix was made using Bfo2291, BFfo2290+ and Bfo2285 at final concentrations of 5mg/ml for each protein. was added to each A 20 μl fraction was taken from each digest and treated with an equivalent volume of cold 95% ethanol stored at -20˚C to precipitate protein and undigested carbohydrate. The samples were briefly centrifuged at 15000 g for 2 minutes, and then dried using a Labconoco SpeedVac Centrivap at 60˚C and 2000 rpm. The lyophilized samples were resuspended in 5 μl of 12.5 mM 2-aminoacridone (AMAC) followed by 5 μl of 1.25M sodium cyanoborohydride then incubated overnight at

37˚C. Samples were dried and resuspended in 25 µl of loading dye (62 mM TRIS-HCl, 0.02% (w/v) bromophenol blue, 10% (w/v) glycerol), then electrophoresed at 25 mA constant amperage through a 28% (w/v) acrylamide native gel. Gels were visualized using a VersaDoc VD4000 imaging system with a UV filter at 405 nm excitation, then analyzed using the BioRad Image Lab program.

### Kinetic assay of Bfo2290+

Determination of enzyme kinetics for Bfo2290+ was determined using methods previously described [53]. The pH dependent assay used relies on the release of $H^+$ ions resulting in a change in pH during S1 type sulfatase-mediated hydrolysis of sulfate esters [54]. Purified Bfo2290+ was buffer exchanged into 100mM MOPS pH 7.5 during the desalting step to maintain a similar pKa to the para-nitrophenol (pNP) pH indicator used in the assay. Digests using 10 mg/ml CSA were diluted using decarbonated MilliQ filtered water to final concentrations ranging from 0.0 to 5.0 mg/ml concentrations. The assay also contained pNP at a final concentration of 0.03 µM and a final concentration of Bfo2290+ of 2 µM. Enzymatic reactions were carried out in triplicate and recorded every 30 seconds for 1 hour using a Cary 50 UV-Vis spectrophotometer at 405 nm in a 1ml quartz cuvette. A standard curve was prepared using titrations of 0.05 M HCl into the solution of 0.03 µM pNP and 1 µM MOPS to determine the concentration of $H^+$ for relating colourmetric change with concentration of $H^+$ to produce a Michaelis-Menten plot.

### Crystallization and X-ray diffraction analysis of Bfo2290 and Bfo2294

Bfo2290 was crystallized at 18˚C by hanging drop vapour diffusion on siliconized glass plates by mixing equal parts (2 µl) protein at a concentration of 20 mg/ml with crystallization solution and 500 µl of crystallization solution as the reservoir solution. The crystallization solution contained 100 mM sodium citrate pH 5.6, 0.5 M ammonium sulfate, and 1 M lithium sulfate. Crystals were harvested for diffraction within 3–4 weeks using 30% (w/v) glycerol as a cryoprotectant. X-ray diffraction data was collected at the Canadian Light Source (CLS) using the Canadian Macromolecular Crystallographic Facility insertion device (CMCF-BM) 081D-1 beamline at 100 K and at a wavelength of 0.98011 Å. Phase data was obtained using molecular replacement with a chondroitin/dermatin sulfate O-4-endosulfatase from marine bacterium (PDBID: 6J66). The structure of Bfo2294 was similarly determined using a mixture of 40% (v/v) Tacsimate pH 7.0 and 11% (v/v) Tacsimate pH 9.0 and 6.5% (v/v) glycerol as the crystallization solution. The solution was mixed in a 2:3 ratio of crystallization to protein solution, 1ml of crystallization solution as the reservoir solution, and a final protein concentration of 14mg/ml. Tacsimate (Hampton Research) contains a mixture of salts including malonic acid, ammonium citrate tribasic, succinic acid, DL-Malic acid, sodium acetate trihydrate, sodium formate, and ammonium tartrate dibasic titrated to the indicated pH.Ham Bfo2291 has been successfully crystallized using a crystallization solution containing 300mM imidazole pH7.8, 2% (v/v) glycerol, and 15% (v/v) ethylene glycol with a reservoir to protein solution ratio of 1:1 and final protein concentration of 15mg/ml. However, only a partial structure was resolved from the X-ray diffraction given the limited resolution obtained.

## Results and discussion

### The Bfo2285-Bfo2295 cluster encodes chondroitin sulfate A degrading enzymes

Analysis of the genome sequence of *T. forsythia* revealed a cluster of genes predicted to encode for a *SusC/D* gene pair (Bfo_2286, Bfo_2287) with a surface glycan binding protein

(Bfo_2289), an enzyme assigned to the family GH88 (Bfo2285), an S1 sulfatase (Bfo2290), and a polysaccharide lyase assigned to the family PL33 (Bfo2291), suggesting this gene cluster could be a PUL specific for GAGs similar to a recently described gene cluster found in *B. thetaiotaomicron* by Ndeh *et al.*. This 11-gene cluster also encodes for 4 additional carbohydrate-related genes (Bfo_2288, and Bfo_2292–2294) and a putative cytoplasmic membrane transporter (Bfo_2295) (Fig 1). Analysis using STRINGv11 also predicted a putative regulatory transcription factor (Bfo_3043) located downstream from the gene cluster which share structural homology to BT3334 [26, 49]. The comparatively fewer number of sulfatases and polysaccharide lyases when compared to the PUL described by Ndeh *et al.* in *B. thetaiotaomicron* may suggest that this gene cluster specifically targets one type of GAG. Using *B. thetaiotaomicron* as a model organism and bioinformatics, the pathway and cellular localization of each protein in the gene cluster are shown in Fig 1.

Based on described pathways, Bfo2291, Bfo2290, and Bfo2285 were predicted to be the first three enzymes in the degradation pathway encoded for by the gene cluster (Fig 2). Activity assays conducted by Eshaque *et al.* suggested that Bfo2291 has a high preference for CSA, which was carried over into research conducted on other proteins from the gene cluster [55]. Activity assays were performed for Bfo2290 and Bfo2285 using the product of the previous enzyme as show in Fig 2, as well as undigested CSA.

FACE was used to assess the enzymatic activity of purified proteins on CSA by visualizing changes in the electrophoretic mobility of the carbohydrate as the molecule is changed by the enzyme (Fig 3). Intensities of spots on the FACE gel were converted into chromatograms for more precise analysis using the BioRad Image Lab program. Due to a series of supply chain related issues, standards were unable to be obtained to estimate the length of the digested fragments. Therefore, structural identification of the analytes and their lengths were unable to be obtained. However, electrophoretic mobility is likely to be influenced by the length and number of sulphate groups attached to the analyte. Undigested CSA was used as a comparative control (Fig 3, lane A). Digestion of CSA with Bfo2291 showed an increase in the number of visible bands suggesting that Bfo2291 may be active as an endo-lyase on CSA (Fig 3, lane B). Digestion with Bfo2290+ on the chondroitin sulfate polymer showed notably decreased fluorescence intensity of bands between the top and bottom most bands, suggesting some activity on CSA in its polymerized state (Fig 3, lane C). Digestion with Bfo2285 showed little if any substantial change to the resulting banding pattern, suggesting that Bfo2285 was not active on the substrate (Fig 3, lane D). When digesting with both Bfo2291 and Bfo2290+, the multiple bands observed in Fig 3, lane B disappeared, and clear separation into 4 major bands was observed (Fig 3, lane E). This may suggest that sulfation has a large effect in the electrophoretic mobility of carbohydrates through the acrylamide gel and could also mean that Bfo2290+ was catalytically active on molecules contain in those bands. The banding pattern for digestion with Bfo2291 and Bfo2285 was very similar to Fig 3, lane B, and showed almost no change, suggesting that Bfo2285 was not active on the still sulfated product of Bfo2291 (Fig 3, lane F). The banding pattern for the digest of Bfo2290+ and Bfo2285 was most similar to the banding pattern in Fig 3, lane C, supporting the hypothesis that Bfo2291 plays a crucial first step in CSA degradation (Fig 3, lane G). While the data supports the proposed model with Bfo2291 acting before Bfo2290+ in the pathway, the banding pattern for the digest containing all three enzymes showed little to no change when compared to Fig 3, lane E, which could suggest that either Bfo2285 was not active, or the dye was unable to bind to the product of this reaction (Fig 3, lane H).

The kinetics of Bfo2290+ were explored using a pH sensitive assay. Michaelis-Menten kinetics of Bfo2290+ was conducted on CSA in its polymeric form as well as the unsaturated CSA product of enzymatic digestion with Bfo2291 (ΔDi-4S), in order to infer substrate

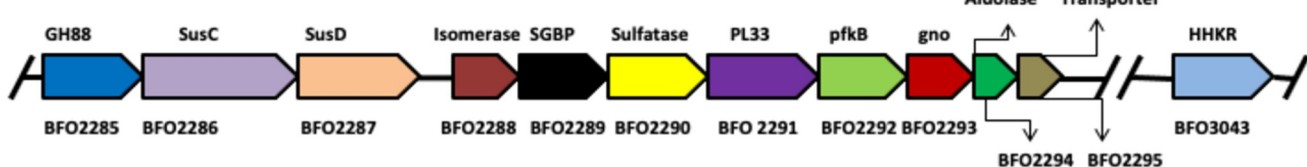

| Gene Tag | Name | Function |
|---|---|---|
| BFO2285 | GH88 | Unsaturated glucuronyl hydrolase |
| BFO2286 | SusC | Ton-B dependent glycan transport |
| BFO2287 | SusD | Glycan binding and acquisition DUF4861 |
| BFO2288 | Isomerase | 4-deoxy-L-threo-5-hexosulose-uronate ketol-isomerase |
| BFO2289 | SGBP | Surface glycan binding protein |
| BFO2290 | Sulfatase | Exo 4-O GalNAc Sulfatase |
| BFO2291 | PL33 | Endo acting chondroitin sulfate A lyase |
| BFO2292 | pfkB | Carbohydrate Kinase |
| BFO2293 | Gno | Gluconate-5-dehydrogenase |
| BFO2294 | Aldolase | Bifunctional 4-hydroxy-2-oxoglutarate / 2-dehydro-3-deoxy-phosphogluconate Aldolase |
| BFO2295 | Transporter | MFS Monosaccharide Transporter |
| BFO3043 | HHKR | Hybrid sensor histidine kinase response regulator |

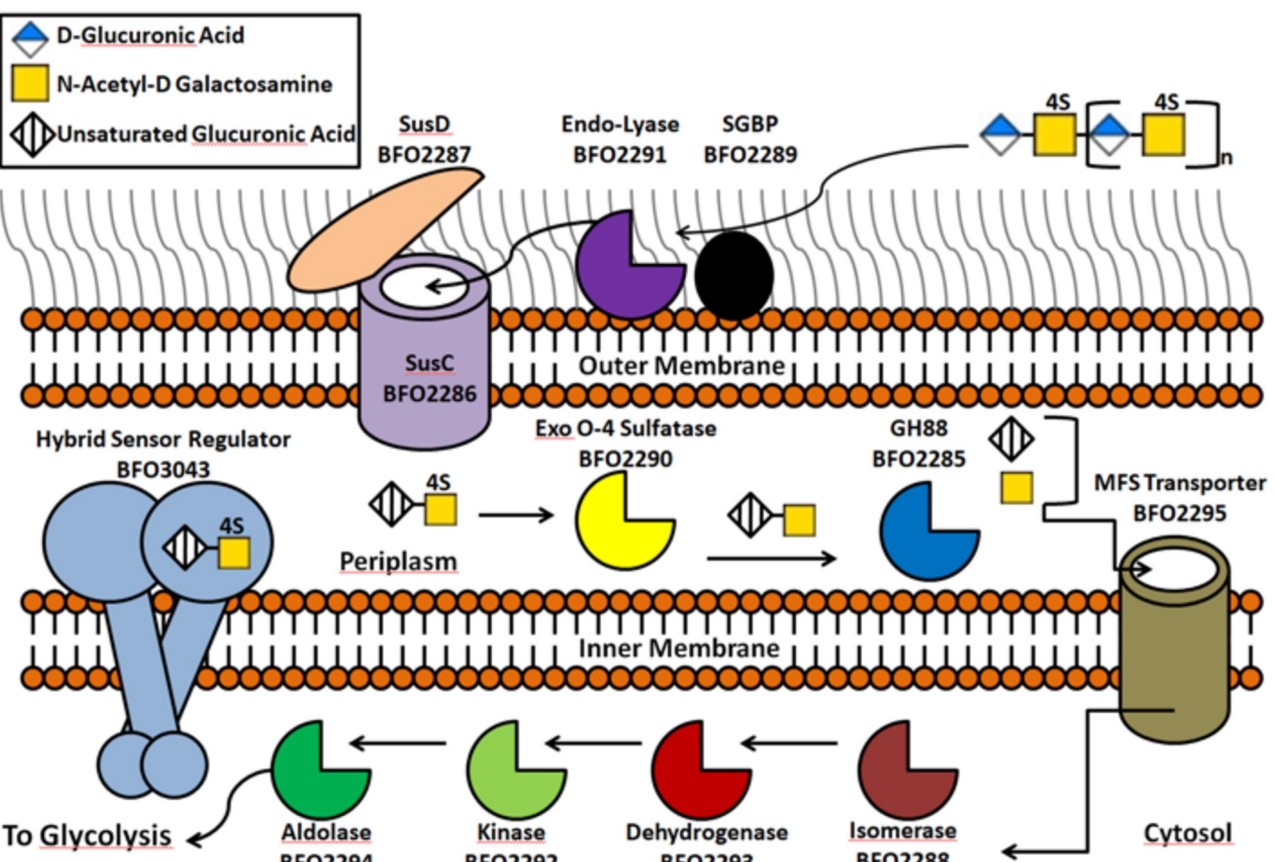

**Fig 1. The putative chondroitin utilization operon and predicted locations of associated gene products from *T. forsythia*.** The gene cluster was identified using DOOR2, FGENESb, and STRINGv11, and the putative function and protein families were determined using InterPro and Phyre2 are summarized. The names given to each protein are representative of their protein families, or their putative function if they were not predicted to be highly homologous to known protein families. Localization was predicted using PSORT and SignalP web servers and depict the path CSA takes as it gets degraded by the operon.

**Fig 2. The hypothesized degradation pathway of chondroitin sulfate A by *T. forsythia*.** Bfo2291 was predicted to be a putative chondroitin sulfate lyase that degrades linear polymers of chondroitin-4-sulfate into its unsaturated disaccharide subunits by cleaving the 1,4 glycosidic linkage between N-acetyl-D-galactosamine and glucuronic acid. Bfo2290 subsequently was hypothesized to be a chondroitin-4-sulfate which removes the O4 sulfate moiety allowing Bfo2285, a putative unsaturated chondroitin hydrolase, to break it down further into its monosaccharide constituents by breaking the 1,3 glycosidic linkage between glucuronic acid and N-acetyl-D-galactosamine.

specificity, and whether Bfo2290+ is active in an endolytic or exolytic capacity. The results of the assay indicate that Bfo2290+ was active on both the CSA substrate as well as ΔDi-4S. A decrease in specific activity when using CSA in its polymeric form suggested that Bfo2290 + was an exo-active protein and only cleaves a sulfate group from the exposed terminal Gal-NAc carbohydrate (Fig 4C). Bfo2290+ was shown to likely be an exo-N-acetylgalactosamine-4-sulfatase with a reported $K_m$ of 0.752 mg/ml ± 0.596 mg/ml, $K_{cat}$ of 3.742 min$^{-1}$ ± 0.878 min$^{-1}$, and $V_{max}$ of 7.48 μM/Min ± 1.76 μM/Min (Fig 4D). $K_{cat}$ was determined with the assumption that all of the Bfo2290+ expressed was post translationally modified, and catalytically active. This assumption likely overestimates the proportion of catalytically active Bfo2290+, and therefore underestimates the value of $K_{cat}$. The substrate concentration was measured in mg/ml due to the varying length of CSA that was used. The kinetic values reported were specific to the method of purification with Bfo2290+, and the percentage of correctly folded protein with post-translational modification could not be determined. The kinetic analysis of Bfo2290 were unable to be obtained due to rapid denaturation and precipitation in the low buffering condition of the assay, suggesting that the formylglycine post-translational modification also increased stability in solution.

The observations made using FACE and the pH dependent kinetic assay suggest that Bfo2290 likely performs the catalytic role described in Fig 2 is strong evidence to suggest that Bfo2290 is an S1 exo-N-acetylgalactosamine-4-sulfatase similar to ones found in the previously described PUL from *B. thetaiotaomicron*, however attempts at characterizing enzyme kinetics for Bfo2285 have thus far been unsuccessful. Enzyme kinetic studies previously conducted by Eshaque *et al.* was successful in showing that Bfo2294 is a KDPG aldolase [55]. As the enzyme was found to be dependent on NADH as a cofactor, the enzyme kinetic constants of Bfo2294 were determined by quantifying the decrease in concentration of NADH by measuring the decrease in absorbance at 340nm as it was consumed in the reaction [55]. Analysis of substrate specificity and pH optimization by Eshaque *et al.* also showed that the preferred substrate for Bfo2294 was CSA at pH 6.5 [55]. The assay was not repeated for KHG, however through multiple sequence alignments and comparisons against structural homologs, Bfo2294 may also be capable of similarly degrading KHG [55].

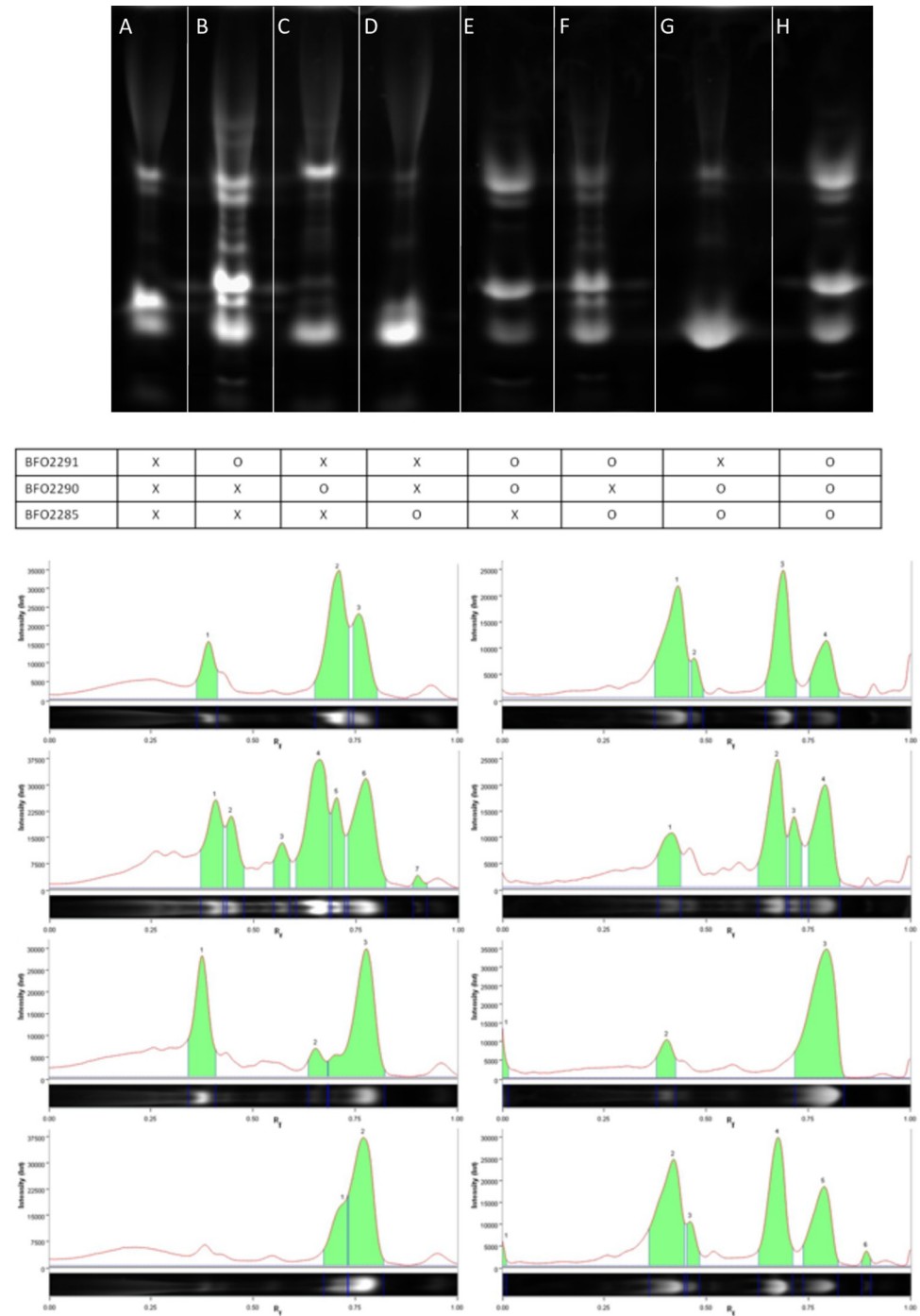

| | A | B | C | D | E | F | G | H |
|---|---|---|---|---|---|---|---|---|
| BFO2291 | X | O | X | X | O | O | X | O |
| BFO2290 | X | X | O | X | O | X | O | O |
| BFO2285 | X | X | X | O | X | O | O | O |

**Fig 3. FACE analysis of chondroitin sulfate A degradation by Bfo2285, Bfo2290, and Bfo2291.** CSA from bovine trachea was digested using each enzyme, then tagged with 2-aminoacridone. The tagged carbohydrates were analyzed using a 28% (w/v) acrylamide gel and suggests Bfo2291 and Bfo229 activity occurs sequentially in CSA degradation. However, Bfo2285 did not show signs of catalytic activity in this assay. The matrix below the acrylamide gel are denoted with an "X" representing an absence of a given protein, and O denoting the presence of a given protein label on the left most edge. The Fig was generated using two separate FACE to maximize lane separation and prevent bleed over events that were likely to occur due to the long running time and running temperature. Gels were color inverted then cropped and the lanes were re-arranged to omit blank lanes in Adobe Photoshop CS6. Refer to S1 Fig for the original blots.

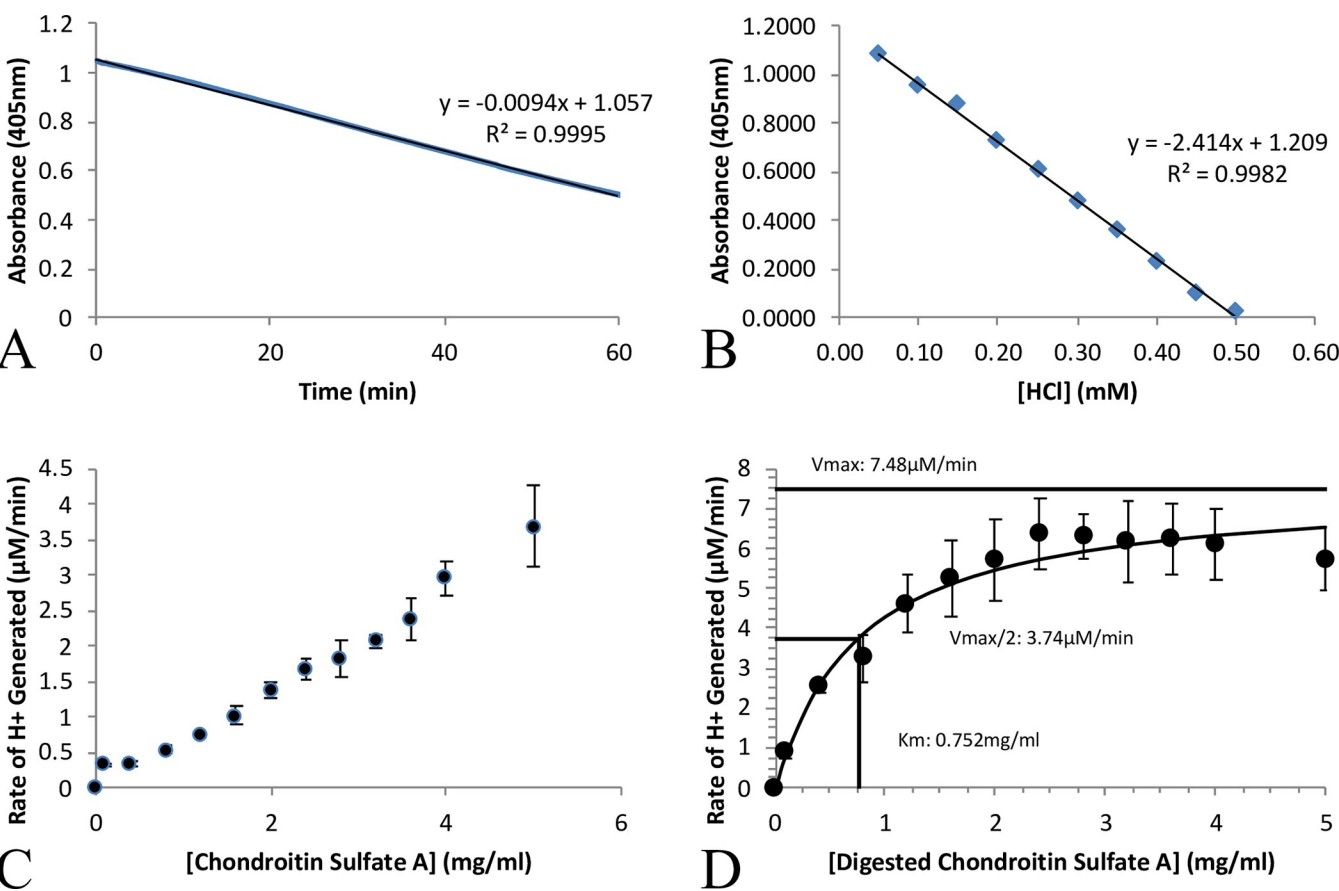

**Fig 4. Michaelis-Menten kinetics of Bfo2290+ on chondroitin sulfate A.** A) The concentration of H$^+$ ions released was determined using a standard curve, and titration with HCl. B) After 1 hour, a decrease in absorbance at 405 nm was observed with 0.6 mM of protein, and 5.0 mg/ml CSA digested by Bfo2291. C) Undigested CSA was unable to reach V$_{max}$ using the same concentration of substrate. D) The reported $V_{max}$ was 7.48 μM/min ± 1.76 μM/Min, $K_{cat}$ of 3.74 min$^{-1}$ ± 0.88 min$^{-1}$, and $K_m$ was 0.75 mg/ml ± 0.60 mg/ml.

## Crystal structure of Bfo2290, Bfo2294 and diffraction of BFO2291

To understand the molecular basis of Bfo2290 function, the native structure was successfully elucidated using X-ray crystallography for the first time and solved using molecular replacement to determine phase information from marine *Vibrio* sp. (PDBID: 6J66) [56]. The structure was solved to a resolution of 2.85 Å in the space group of P2$_1$2$_1$2$_1$ and iteratively built and refined to a corresponding R$_{work}$ of 0.230 and an R$_{free}$ of 0.276 (Table 1). The refined model contained 3 protein molecules in the asymmetric unit arrange as a trimer with C3 symmetry (Fig 5A). The structure beginning from the N-terminal domain started at Arg31 and was modeled up to Asp511. 30 residues were missing from the N-terminal domain, and 5 residues were missing from the C-terminal domain. Residues Thr180-Arg178, Gly189-His196, and Asp425-Gly428 were also removed due to poor electron density (Fig 5B). The model contained an α/β topology shown using a topological diagram of the polypeptide fold with 13 β-strands (β1-β13), and 16 α-helices (α1-α16) (Fig 5C). Two distinct domains were observed with the larger one containing a 9-strand parallel β sheet (N-terminal domain) with the exception of β4 and β8 which were anti-parallel. The smaller second domain (C-terminal domain) contained a beta sheet consisting of 4 anti-parallel beta strands (β10-β13) and a long alpha helical region (α16) running orthogonal to the β strands but in the same plane as a solvent exposed structure.

**Table 1. X-ray data collection and structure statistics for Bfo2291, Bfo2290, and Bfo2294.**

| Data Collection | Bfo2291 | Bfo2290 | Bfo2294 |
|---|---|---|---|
| Beamline | CHESS Bio-MX | CMCF-BM 08ID-1 | CMCF-BM 08ID-1 |
| Resolution (Å) | 30–3.34 (3.53–3.34) | 48–2.85 (3.00–2.85) | 43–2.20 (2.32–2.32–2.20) |
| Space Group | I4 | $P2_12_12_1$ | $P2_3$ |
| Cell Dimensions | | | |
| $a, b, c,$ (Å) | 189.12, 189.12, 138.35 | 113.18, 121.11, 144.66 | 98.15 |
| $\alpha, \beta, \gamma,$ (°) | 90 | 90 | 90 |
| Wavelength (Å) | 1.12710 | 0.98011 | 0.97948 |
| Measured Reflections | 478475 | 630031 | 179533 |
| Unique Reflections | 34776 | 47085 | 16307 |
| Completeness | 98.7 (92.2) | | 99.9 (99.9) |
| $R_{merge}$ | 0.128 (1.008) | 0.200 (1.905) | 0.135 (0.567) |
| Multiplicity | 13.8 (13.0) | 13.4 (14.0) | 11.0 (7.2) |
| I / σI | 11.9 (3.1) | 12.7 (1.6) | 12.6 (3.3) |
| $CC_{1/2}$ | 0.999 (0.915) | 0.997 (0.599) | 0.996 (0.871) |
| **Refinement** | | | |
| Resolution (Å) | | 2.85 | 2.20 |
| $R_{work}/R_{free}$ | | 0.222/0.281 | 0.179/0.221 |
| **No. Atoms** | | | |
| Protein | | 10749 | 1656 |
| Solvent | | 24 | 89 |
| **B-Factors** | | | |
| Protein | | 72.9 | 38.4 |
| Solvent | | 63.4 | 44.4 |
| **RMSD** | | | |
| Bond lengths (Å) | | 0.003 | 0.003 |
| Bond angles (°) | | 0.730 | 0.610 |
| **Ramachandran** | | | |
| Preferred (%) | | 95.6 | 98.2 |
| Allowed (%) | | 3.8 | 1.8 |
| Disallowed (%) | | 0.6 | 0 |

Many of the conserved residues in the active site which contributed to the geometry of the active site were also conserved among the 5 structures compared from the multiple sequence alignment such as: D40, N112, K145, H147, H248, D326, and K340 (Fig 6A). Some key residues that also may have contributed to the geometry of the binding site on BT3349 that were not modeled in Bfo2290 due to the exclusion of residues from the crystal structure were H140, W163, T183 and D185 (on the BT3349 structure). Topologically Bfo2290 was very similar to BT3349 from *B. thetaiotaomicron* [26]. To compare conservation of the active site and evaluate positions of key catalytic residues, the structure of Bfo2290 was superimposed on to the structure of BT3349 using USFC Chimera and the most highly conserved residues were highlighted using a multiple sequence alignment of the best five homologous structures found using Phyre2 [44]. The superimposed structures had a final alignment root mean squared deviation (RMSD) of 1.105 Å, structural distance measure (SDM) of 11.021 with a cutoff of 10, and a Q-score of 0.797 (Fig 6B). Overall, 36 residues were conserved across all protein structures, with the exception of Y416, F565, and D469, all were found in the N-terminal domain. Many of the residues found in the active site were positively charged amino acids, which may contribute to substrate binding of the negatively charged chondroitin sulfate species. Further investigation

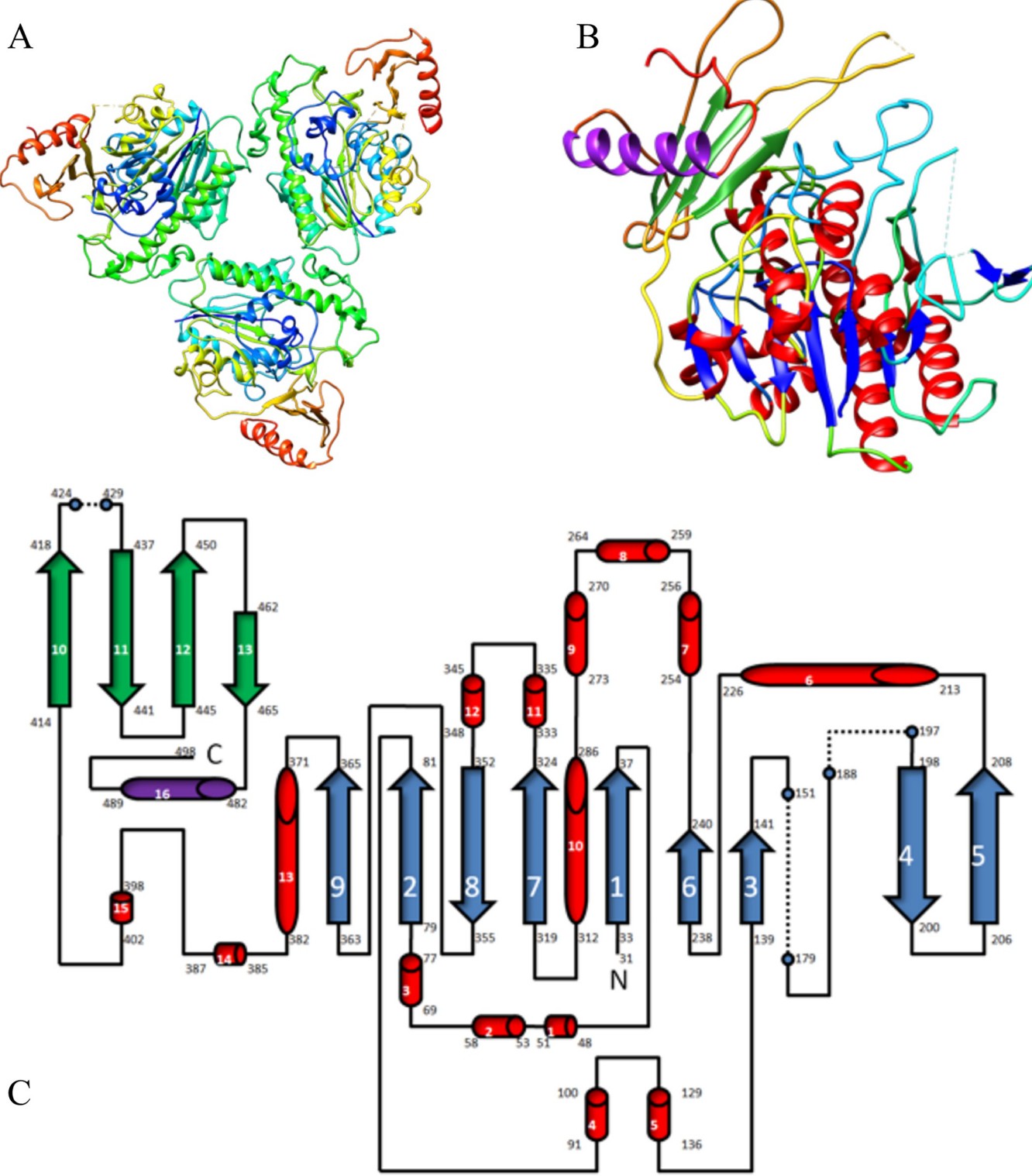

**Fig 5. Crystal structure of Bfo2290 arylsulfatase.** The model was refined with a resolution of 2.85Å, in the P2$_1$2$_1$2$_1$ space group. A) The asymmetric unit contained 3 proteins arranged in a trimer with C3 symmetry. B) A single chain in the asymmetric unit models a monomer and has an α/β topology typical of glycosaminoglycan degrading sulfatase enzymes. A ribbon structure highlighted secondary structural elements. C) A topology diagram of the structure of Bfo2290 was made with secondary structures highlighted in colours matching the single chain model. Dotted lines represented residues that were removed due to insufficient data.

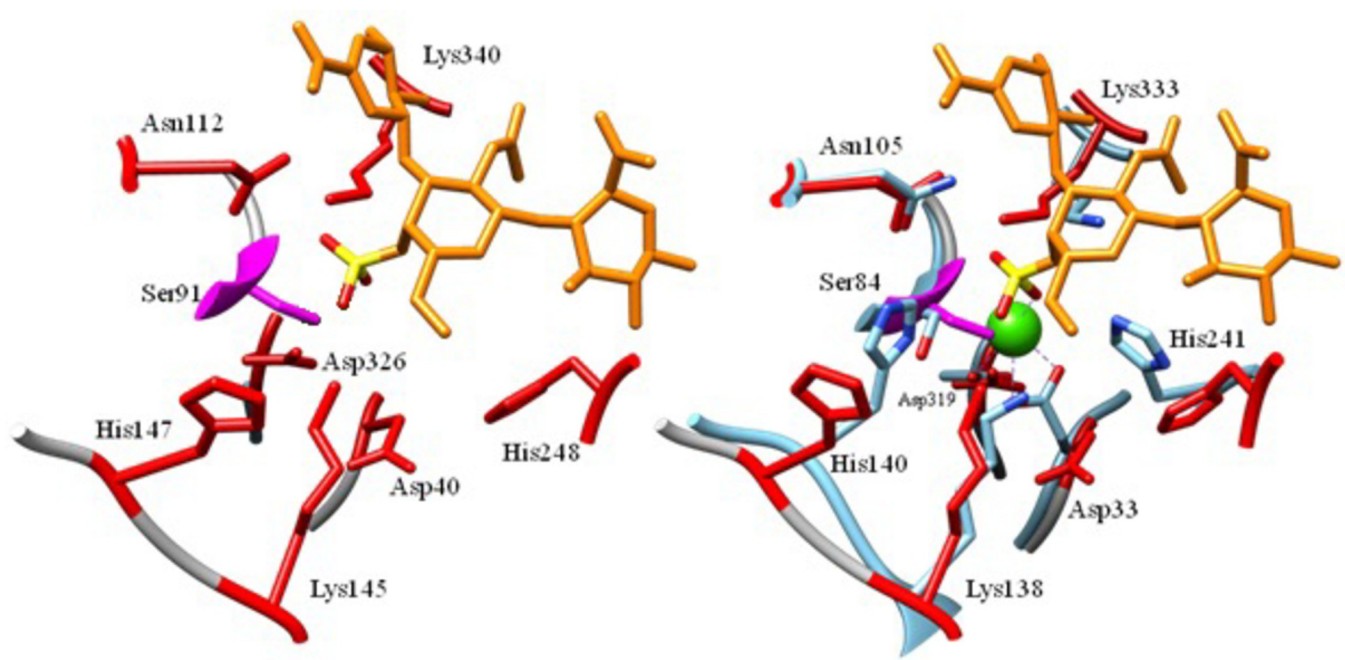

**Fig 6. Conserved active site residues in Bfo2290 and structurally similar proteins, and superimposition of BT3349.** A) Conserved residues highlighted in red represent residues that were conserved among 5 sulfatases that are structurally similar and had high sequence identity to Bfo2290. The serine residue post translationally modified into formylglycine was highlighted in pink. The superimposed substrate from BT3349 was highlighted in orange, was a trisaccharide of CSA containing 1 sulfated N-acetyl-D-galactosamine flanked on either side by glucuronic acid. B) Bfo2290 was superimposed on to the crystal structure of BT3349 which shares 61% sequence identity and a highly similar structure. Residues labeled are from BT3349.

of the active site residues using site-directed mutagenesis (SDM) would be required using the residues described above as potential candidates.

The structure of Bfo2294 and partial structure of Bfo2291 was previously described by Eshaque *et al.* and reported here as part of the CSA specific PUL-like operon in *T. forsythia* [55]. The structure of Bfo2294 was solved to a resolution of 2.20Å in the space group of P2₃ and iteratively built and refined to a corresponding $R_{work}$ of 0.179 and $R_{free}$ of 0.221 (Table 1). Bfo2294 was found to have an α/β-barrel structure and contained a glutamic acid and lysine zwitterionic-pair typical of class I aldolases which use a Schiff-base mechanism to convert KDPG and KHP into D-glyceraldehyde-3-phosphate and pyruvate (Fig 7) [55].

## Implications of the structural and functional characterizations of Bfo2285, Bfo2290, and Bfo2291

The *T. forsythia* PUL-like operon encodes for proteins that perform key functions of a classical PUL, such as a SusC/D-like surface transport channel (Bfo_2286 and Bfo_2287), a transmembrane hybrid sensor regulatory protein, as well as what is suspected to be a monosaccharide import channel. While the PUL described by Ndeh *et al.* was demonstrated to be capable of metabolizing a diverse range of glycosaminoglycans, the *T. forsythia* operon like other species that diverged from the same evolutionary ancestor, was shown here to be more concise; expressing fewer proteins that have a more specific protein functions related to GAG-utilization. Although much of the results proposed regarding the model of the system in *T. forsythia* is primarily based on bioinformatics, given the evolutionary lineage and metabolic specificity of similarly divergent but more closely related species, there is a strong speculative case to support this model. Furthermore, given the high structural similarity of Bfo2290 to its functional

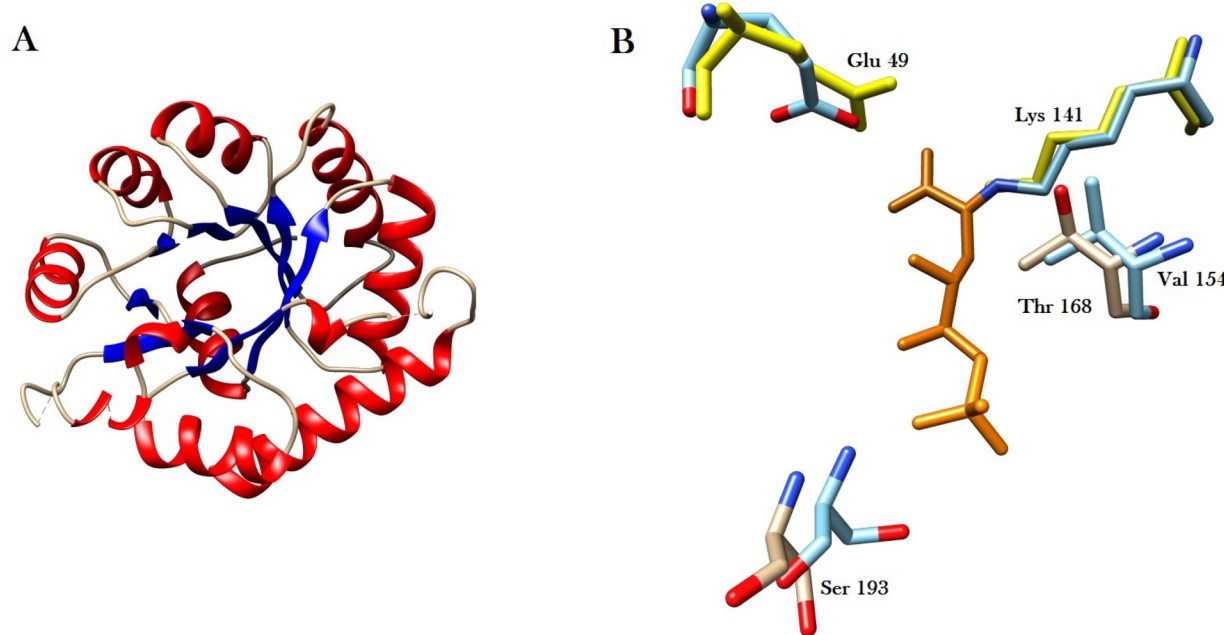

**Fig 7. The crystal structure of Bfo2294.** A) Bfo2294 shows an α/β-barrel tertiary structure. B) The active site contains conserved residues modeled against a KDPGal aldolase from *E. coli* (PDBID: 2V82). Key residues highlighted in yellow Glu 49 and Lys 141 form a zwitterionic-pair critical to the catalytic function of Bfo2294. KDP was modeled in orange as part of the co-crystal structure of a KDPGal aldolase to show the orientation of the substrate in the active site.

and structural homolog from *B. thetaiotaomicron* (BT3349), there is further evidence to support that *T. forsythia* contains a PUL-like operon that has evolved for CSA specificity.

Analysis by FACE further demonstrates the stepwise degradation of CSA by Bfo2291 then Bfo2290, although further analysis of the degradation products may be necessary to further describe the similarities between *T. forsythia* and *B. thetaiotaomicron*. While FACE analysis was conducted without standards and did not provide the qualitative identification of specific products, it did reveal changes to the composition of the CSA solution in a qualitative way, as well as relative quantitative ratios of the different enzyme products in the solution. Digestion of the starting material, which contained 3 major peaks forming 2 separate clusters (shown in Fig 3A), with Bfo2291 produced 7 peaks in the chromatogram, suggesting multiple products and endolytic degradation of CSA (Fig 3B). Digestion with Bfo2290+ and Bfo2291 suggested that while Bfo2290+ was capable of digesting CSA as shown by the change in banding pattern and chromatogram (Fig 3F), a shift towards a more homogenous end-product and fewer peaks suggest that the combination of Bfo2290+ and Bfo2291 is more efficient at degrading CSA (Fig 3D). Therefore, it is likely that the combination of Bfo2290+ and Bfo2291 produces an unsulfated and unsaturated disaccharide derivative of CSA. For reasons currently unclear, this would also further support the findings of the kinetic assay, which suggests that Bfo2290+ is an endolytic arylsulfatase. This could suggest that Bfo2291 could be an excreted enzyme, which implies that the Bfo2286/Bfo2287 SusCD channel is selective for disaccharides of CSA, however there is currently no evidence to support this. FACE analysis also suggested some catalytic activity was present in Bfo2285, however kinetic studies were unable to show signs of catalytic function.

Kinetic characterization of Bfo2290 showed activity and substrate specificity towards the unsaturated disaccharide CSA product of Bfo2291 which suggests an exolytic mechanism. The

kinetic characterization of Bfo2290 not only showed that kinetic rates similar to other arylsulfatase proteins, but the reaction continued at a linear rate through the entire hour length of the assay, suggesting that Bfo2290 was still able to eliminate the sulfate group from the active site and regenerate the aldehyde. Therefore, it may be worth some investigation in the future as to whether this meant that the post-translational modification of Cys91 to formylglycine is critical for calcium incorporation, or Bfo2290 eliminates the need for a calcium cation all together. While it is currently unclear how the mechanism may function without a calcium cation as the crystal structure does not appear to contain a suitable replacement for calcium that would be occupied. Attempts to increase protein production and create a better stabilizing buffer by adding calcium salts to the growth media and lysis buffer also did not appear to noticeably improve either protein stability or yield when compared to protein isolated using the methods described here.

Based on the crystal structure and high degree of similarity to homologous enzymes, Bfo2290 presents an interesting exception to the general mechanism of formylglycine dependent arylsulfatases. It is generally understood that arylsulfatases hydrolyze the sulphur-ester bond using a two-step ping pong mechanism [57, 58]. The general mechanism described by Boltes *et al.* using an arylsulfatase from *Pseudomonas aeruginosa* highlights the role of formylglycine and a calcium cation typical of arylsulfatases [58]. The formylglycine residue as found in Bfo2290 and other arylsulfatases is recognized as being a Cα-formylglycine hydrate in the resting state of the enzyme which forms a geminal diol [58]. One of the alcohol groups establishes a hydrogen bond while the other is oriented close enough to the sulfate sulfur atom to start a nucleophilic attack that forms a sulfoenzyme intermediate [58]. The role of calcium in this mechanism is for the orientation of the substrate as it occupies the active site [58]. The calcium cation helps stabilize one of the alcohol groups on the formylglycine residue and is also key to removing the sulfate after the nucleophilic attack to regenerate the aldehyde by shifting the equilibrium of the sulphated aldehyde and hydrated aldehyde towards the hydrate [58]. The crystal structure of Bfo2290 did find evidence of a calcium cation however, it is important to note that the protein used to crystallize Bfo2290 did not contain the post-translational modification of Ser91 to formylglycine which could play a role in the incorporation of the calcium cation.

The active site residues of Bfo2290 are highly conserved, and the overall residues that delineate the active site share high structure and sequence homology with BT3349. When the binding pocket surface was modeled for Bfo2290, many of the residues were found to be highly conserved even amongst other similar GAG degrading sulfatases (PDBID: 6S21, 6J66, 2QZU, 5G2V, and 6PRM), and many additional residues were found to be conserved between Bfo2290 and BT3349. When Bfo2290 and BT3349 were superimposed, it was found that many of the residues found in the binding pocket were positively charged residues or polar nitrogen containing residues with a positive dipole facing the substrate modeled in the BT3349 co-crystal structure. As CSA is generally regarded to be negatively polarized, this may indicate that binding affinity may be attributed to the many positive surface residues of Bfo2290 and BT3349 which form electrostatic interactions with the substrate. Although current experimental evidence supports an exolytic catalytic mechanism, or a high preference for ΔDi-4S, the only major structural difference noted between Bfo2290 and BT3349 (an endolytic sulfatase), is the absence of a bound calcium cation metal cofactor.

## Conclusions

Based on genomic and bioinformatics analysis there is a high degree of likelihood that *T. forsythia* expresses a PUL-like operon that is descendent of *B. thetaiotaomicron* for degrading

complex carbohydrates such as CSA. As CSA is the primary GAG in alveolar bone, this likely serves as a mechanism by which *T. Forsythia* compromises periodontal tissues contributing to bone loss in periodontal disease in conjunction with host osteoclastic activity. Efforts to characterize the proteins in this PUL-like operon have produced two crystal structures for Bfo2290 and Bfo2294 which both share a high degree of structural similarity to other well described structural homologs in their respective families. Kinetic assays of both Bfo2294 by Eshaque *et al.* and the kinetic assay of Bfo2290 described here suggest that the proteins are active on CSA with Bfo2290 being most active on the ΔDi-4S substrate produced using Bfo2291. The average $V_{max}$ was measured to be 7.48 μM/min ± 1.76 μM/Min, the average $K_{cat}$ of 3.74 min$^{-1}$ ± 0.88 min$^{-1}$, and the average $K_m$ was determined to be 0.75 mg/ml ± 0.60 mg/ml for Bfo2290. The crystal structure of Bfo2290 which was refined to a resolution of 2.85Å using X-ray crystallography revealed a peculiar lack of dependence on a divalent calcium cation cofactor which may suggest *T. forsythia* has evolved in a divergent manner to utilize a divergent catalytic mechanism from other arylsulfatases. Analysis by FACE and the kinetic assay of Bfo2290 reported herein indicate that Bfo2291 is likely an endolytic lyase while Bfo2290 is likely an exolytic arylsulfatase which disconcerting with regards to the cellular localization of Bfo2291 in the model suggested here. While Bfo2291 may be an excreted enzyme, the exact compartmentalization remains to be determined. Further analysis of the PUL-like operon from *T. forsythia* through structural analysis of the various proteins described in Fig 1 will be necessary to further understand how CSA is degraded and utilized by *T. forsythia*. While further analysis of the products of Bfo2291 and Bfo2290 through mass spectrometry as well as cellular localization studies and expression studies through fluorescently labeled cell localization studies and RT-PCR were aspirational goals for reporting with the crystal structure, time constraints under a global pandemic proved this to be a difficult endeavor. Despite this, the results thus far may have significant implications with regards to our understanding of periodontal disease as well as the genetic lineage of *T. forsythia*. With the evidence thus far described in this paper, propose to expand the current database of known PULs to include the system described here.

## Supporting information

**S1 Fig. The original unmodified gel image was used for the creation of Fig 3 in the manuscript.** Gel 1 was used in the creation of Fig 3A–3D, and gel 2 was used in the creation of Fig 3E–3H. The gel was loaded from left to right in the annotated order. Experimental samples are labelled using the relevent figure subheading and coloured blue. Blank lanes are labelled X in yellow. Lanes were cropped, rearranged to omit blank lanes, and color inverted using Adobe Photoshop CS6 in order to create the figure.
(PDF)

**S1 Table. Truncated amino acid sequences of Bfo2285, Bfo2290, and Bfo2291 used for recombinant expression in *E. coli*.** Amino acids highlighted in red were removed in the truncation, while amino acids highlighted in green were recombinantly expressed.
(DOCX)

**S2 Table. Primers for PCR amplification of Bfo2285 from pET21b plasmid for ligation into pET28a.**
(DOCX)

## Acknowledgments

The X-ray diffraction models and datasets presented in this study can be found at the Protein Data Bank: accession numbers 8DI0 and 8DI1 for Bfo2290 and Bfo2294, respectively. Datasets

were collected at the Cornell High Energy Synchrotron Source beamline ID7B2 (Bfo2291) and the Canadian Light Source beamlines CMCF-BM and BMCF-ID (Bfo2290 and Bfo2294). We are very grateful for the work done by Michel Fodje of the CLS to assist us with early data collection and processing of twinning and overlapping lattice issues observed for many Bfo2294 datasets. After our manuscript submission a comprehensive characterization of mucin-degrading sulfatases was reported by Luis et al. (PMID 34616040).

## Author Contributions

**Conceptualization:** Michael D. L. Suits.

**Data curation:** Anthony Dang, Michael D. L. Suits.

**Formal analysis:** Peter Nguyen, Rony Eshaque, Michael D. L. Suits.

**Funding acquisition:** Michael D. L. Suits.

**Investigation:** Rony Eshaque, Barbara Anne Garland, Anthony Dang, Michael D. L. Suits.

**Methodology:** Rony Eshaque, Barbara Anne Garland, Anthony Dang, Michael D. L. Suits.

**Project administration:** Barbara Anne Garland, Michael D. L. Suits.

**Software:** Michael D. L. Suits.

**Supervision:** Michael D. L. Suits.

**Validation:** Peter Nguyen, Michael D. L. Suits.

**Visualization:** Peter Nguyen, Michael D. L. Suits.

**Writing – original draft:** Peter Nguyen, Michael D. L. Suits.

**Writing – review & editing:** Peter Nguyen, Michael D. L. Suits.

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
