## [Decision Letter · Decision Letter 0]

28 Dec 2021

PONE-D-21-34850Degradation of Chondroitin Sulfate A by a PUL-like operon in Tannerella forsythiaPLOS ONE

Dear Dr. Suits,

Thank you for submitting your manuscript to PLOS ONE. After careful consideration, we feel that it has merit but does not fully meet PLOS ONE’s publication criteria as it currently stands. Therefore, we invite you to submit a revised version of the manuscript that addresses the points raised during the review process.

We look forward to receiving your revised manuscript.

Kind regards,

Nikos K Karamanos, Ph.D.

Academic Editor

PLOS ONE

Journal Requirements:

“PN, BG, RE, and AD were supported by studentships provided by Wilfrid Laurier University. This work was supported by a National Science and Engineering Research Council of Canada Discovery Development Grants (RPGIN-2014-05018 and DDG-2020-00007). Additional support was provided by GlycoNet Grant AM-21. We are grateful for the support of the Canadian Light Source and Cornell High Energy Synchrotron Source where diffraction data were collected.”

6. Please ensure that you refer to Figure 6 in your text as, if accepted, production will need this reference to link the reader to the figure.

Reviewers' comments:

Reviewer's Responses to Questions

**Comments to the Author**

1. Is the manuscript technically sound, and do the data support the conclusions?

Reviewer #1: Yes

Reviewer #2: Yes

2. Has the statistical analysis been performed appropriately and rigorously? 

Reviewer #1: Yes

Reviewer #2: Yes

3. Have the authors made all data underlying the findings in their manuscript fully available?

Reviewer #1: Yes

Reviewer #2: Yes

4. Is the manuscript presented in an intelligible fashion and written in standard English?

Reviewer #1: Yes

Reviewer #2: Yes

5. Review Comments to the Author

Reviewer #1: The aim of the paper by Nguyen et al. was to identify a new polysaccharide utilization locus (PUL) like operon from Tannerella forsythia one of the bacterial triad causing advanced periodontitis. Cristallization data are reported from three proteins encoded by this new PUL-like operon: Bfo2290, Bfo2291 and Bfo2294. Digestion assays using chondroitin sulfate A demonstrate that Bfo2291 and Bfo2290 are a chondroitin sulfate A lyase and a ΔDi-4S sulfatase, respectively. Data from this paper further expands the database of PULs involved in periodontal disease.

The experiments are described in sufficient details and the approaches used to support the results are up to date.

SPECIFIC COMMENTS:

The results provided are convincing; there are some inaccuracies that should be checked.

Materials and Methods session.

Page 9. Line 14 and page 10 line 2. Please check the FPLC equipment, ATKA or AKTA?

Page 10. line 13. Please check the 2-aminoacridone concentration used to label sugars.

Reviewer #2: Authors here present that chondroitin sulfate A may be digested by the enzymes Bfo2290 and Bfo2291 produced by Tanerella forsythia that belong to the PUL-like operon Bfo2285-Bfo2295. The manuscript is well written and the data very interesting. Conclusions support the results. There are some minor comments to concern:

1. In the Introduction, it is better to take away the words "possible" and "frequently" in the phrases "GAGs are a possible carbohydrate component of proteoglycans..." and "GAGs are frequently composed of repeating disaccharide subunits..." as GAGs do compose the glycan part of proteoglycans (this is not a possibility) and always contain repeating disaccharides, not frequently.

2. In Materials and Methods, in the description of FACE analysis at the 3rd lane, please cite the enzymes that you used and the units required for the digestion of Chondroitin sulfate A.

3. In Results, in FACE analysis figure 3, lane A presents the chondroitin sulfate A. There are three bands presented here. In the next lanes, authors support that Bfo2291 digests this GAG. Could you please show in the figure of the bands what kind of GAGs are likely expected to be? Are they long chains, disaccharides or oligo- mono-saccharides? It is not clear the kind of poly-, oligo- or mono-mers or, alternatively, the molecular weight of the resulted bands.

6. PLOS authors have the option to publish the peer review history of their article (what does this mean?). If published, this will include your full peer review and any attached files.

Reviewer #1: No

Reviewer #2: No

---

## [Author Response · Author response to Decision Letter 0]

22 Jul 2022

We are grateful for the work done by the Reviewers, and have incorporated the following changes based on their comments.

Reviewer 1 Responses:

• Revised ATKA to ÄKTA.

• 2-aminoacridone changed to 12.5mM from 12.5M

Reviewer 2 Responses:

• Removed words “frequently” and “possible” throughout

• Materials and methods now include protein identities and concentrations used for digestion

• FACE analysis, quantification and standards unavailable. The experiment was conducted to observe 1) how many bands of product were produced by BFO2291 2) how many bands of product were generated from BFO2290 3) BFO2285 active? Without standards it is inappropriate to assign products to certain bands. Standards are difficult to obtain for comparison, but we are careful not to draw specific conclusions from this analysis. Attempts at conducting Mass Spec were inconclusive, likely due to the heterogeneity of the source of chondroitin A.

---

## [Editor Report · Decision Letter 1]

1 Aug 2022

Degradation of chondroitin sulfate A by a PUL-like operon in Tannerella forsythia

PONE-D-21-34850R1

Dear Dr. Suits,

We’re pleased to inform you that your manuscript has been judged scientifically suitable for publication and will be formally accepted for publication once it meets all outstanding technical requirements.

Kind regards,

Nikos K Karamanos, Ph.D.

Academic Editor

PLOS ONE
---

## [Editor Report · Acceptance letter]

31 Aug 2022

PONE-D-21-34850R1 

Degradation of chondroitin sulfate A by a PUL-like operon in *Tannerella forsythia*

Dear Dr. Suits:

I'm pleased to inform you that your manuscript has been deemed suitable for publication in PLOS ONE. Congratulations! Your manuscript is now with our production department. 

Kind regards, 

on behalf of

Prof. Dr. Nikos K Karamanos 

Academic Editor

PLOS ONE